# New Horizons in the Genetic Etiology of Systemic Lupus Erythematosus and Lupus-Like Disease: Monogenic Lupus and Beyond

**DOI:** 10.3390/jcm9030712

**Published:** 2020-03-05

**Authors:** Erkan Demirkaya, Sezgin Sahin, Micol Romano, Qing Zhou, Ivona Aksentijevich

**Affiliations:** 1Schulich School of Medicine & Dentistry, Department of Paediatrics, Division of Paediatric Rheumatology, University of Western Ontario, London, ON N6A 5W9, Canada; micol.dr.romano@gmail.com; 2Van Training and Research Hospital, Department of Paediatric Rheumatology, 65000 Van, Turkey; sezgin@istanbul.edu.tr; 3Department of Pediatric Rheumatology, ASST-PINI-CTO, 20122 Milano, Italy; 4Life Sciences Institute, Zhejiang University, Hang Zhou 310058, China; zhouq2@zju.edu.cn; 5Inflammatory Disease Section, National Human Genome Research Institute, Bethesda, MD 20892, USA; aksentii@mail.nih.gov

**Keywords:** systemic lupus erythematosus, lupus, SLE, genetic, monogenic, mendelian, familial, complement deficiency, interferonopathies, interferon-stimulated genes (ISGs)

## Abstract

Systemic lupus erythematosus (SLE) is a clinically and genetically heterogeneous autoimmune disease. The etiology of lupus and the contribution of genetic, environmental, infectious and hormonal factors to this phenotype have yet to be elucidated. The most straightforward approach to unravel the molecular pathogenesis of lupus may rely on studies of patients who present with early-onset severe phenotypes. Typically, they have at least one of the following clinical features: childhood onset of severe disease (<5 years), parental consanguinity, and presence of family history for autoimmune diseases in a first-degree relative. These patients account for a small proportion of patients with lupus but they inform considerable knowledge about cellular pathways contributing to this inflammatory phenotype. In recent years with the aid of new sequencing technologies, novel or rare pathogenic variants have been reported in over 30 genes predisposing to SLE and SLE-like diseases. Future studies will likely discover many more genes with private variants associated to lupus-like phenotypes. In addition, genome-wide association studies (GWAS) have identified a number of common alleles (SNPs), which increase the risk of developing lupus in adult age. Discovery of a possible shared immune pathway in SLE patients, either with rare or common variants, can provide important clues to better understand this complex disorder, it’s prognosis and can help guide new therapeutic approaches. The aim of this review is to summarize the current knowledge of the clinical presentation, genetic diagnosis and mechanisms of disease in patents with lupus and lupus-related phenotypes.

## 1. Introduction

Systemic lupus erythematosus (SLE; OMIM 152700) is a complex autoimmune disease not only from a clinical standpoint but also from genetic and immunological aspects of the disease. Multiple studies have shown that loss of immune tolerance to cell surface and nuclear antigens contribute to this disease, however, the underlying etiology and triggering event for impaired immune tolerance remain elusive. 

Up to 100 susceptibility loci for polygenic, multifactorial SLE and more than 30 genes causing the monogenic form of SLE and SLE-like phenotypes have been described, yet they capture only a small proportion of heritability to lupus and other autoimmune diseases (Table 1) [1,2,3,4,5,6]. Discovery of novel or rare disease-causing gene variants in patients with early-onset SLE phenotype has had a significant contribution to the unraveling of molecular pathways involved in the disease pathogenesis [7,8,9]. The term monogenic lupus has been introduced to denote SLE patients who carry high-penetrance either dominantly or recessively inherited pathogenic variants in a single gene. Among these rare forms of the disease, mutations in the genes encoding for complement pathway are the first and foremost described. Apart from the complement deficiencies, the vast majority of the single gene defects that lead to monogenic lupus are categorized under the umbrella of type I interferonopathies [8,9]. This term is broadly defined but essentially indicates persistent upregulation of type I interferon (IFN) signaling as measured by high expression of interferon-stimulated genes (ISGs). Whether this signature is causal to the pathology or represents an association remains unclear and mechanisms leading to the upregulation of interferon signaling are yet to be clarified. The close association between SLE and up-regulated type I IFN signaling has been a well-known entity for a long time [10]. High expression of ISGs (interferon-stimulated genes; also called interferon signature), which is accepted as an indirect and feasible evidence of excessive type 1 IFN production, was invariably found to be higher in blood samples of lupus patients [11]. Type I interferonopathies are considered in the spectrum of autoinflammatory and autoimmune diseases in which the engagement of adaptive and innate immune cells occurs (Figure 1). As a result, patients may exhibit the clinical and molecular features of autoimmune and autoinflammatory disease [12,13,14]. Although, the intricate connections of innate and adaptive immune systems have been studied both in humans and mice, they are still poorly understood. 

The wide range of molecular pathways identified by genetic studies disclose the complexity of hereditary predisposition to lupus. Clinical variability of the disease is further influenced by a number of contributing elements including genetic modifiers, epigenetic and environmental factors. These factors create a barrier to establishing early diagnosis and carrying out appropriate treatment and effective disease monitoring. Furthering our understanding of the genetic causes of lupus and molecular diagnosis are the major steps towards precision medicine in SLE. 

## 2. Genes and Molecular Pathways Associated with Lupus Phenotype

The concordance rate for SLE in monozygotic twins was found to be 24%, compared to only 2% in dizygotic twins [15]. Familial aggregation of various autoimmune diseases including SLE have been demonstrated in a large cohort of Latin American SLE patients and in other founder populations [16]. Compared to adult-onset SLE, patients with childhood-onset SLE are more likely to carry novel/rare high-penetrance variants associated with mendelian lupus or may have a burden of low-penetrance common SLE susceptibility alleles that can have additive effects [7,17]. The high degree of concordance rate among monozygotic twins, substantial familial aggregation and discovery of numerous common and rare variants are all supportive of a strong genetic background for SLE.

It was previously thought that the relevant genetic variants and the causal molecular pathway involved in lupus were associated with distinct clinical presentations. This simplification greatly underestimated the biological complexity of this disease. Mutations within the same gene and identical gene variants have been associated with different clinical phenotypes including lupus, type I interferonopathy or familial chilblain lupus (FCL) (chilblain lupus 1: OMIM 610448; chilblain lupus 2: OMIM614415). Conversely, a significant clinical and molecular overlap has been observed between patients with mutations in different genes. In this review, the genes involved in monogenic lupus will be categorized in four main groups based on the relevant pathways which they are associated with: complement pathway, type I IFN pathway, self-tolerance pathway, and other yet uncharacterized pathways (Table 1).

**Table 1 jcm-09-00712-t001:** Genes linked to monogenic lupus and monogenic immune dysregulations with features of autoimmunity.

Pathway	Gene	Inheritance	Disease	Reference
Complement pathway	*C1q*	AR	SLE, AGS	[18,19,20,21,22,23,24]
	*C1r/C1s*	AD		[7,25,26]
	*C2*	AR		[27]
	*C4*	AR		[28,29,30,31,32]
Type I IFN pathway	*DNASE1*	AD	HUV, lupus	[33,34,35]
	*DNASE2*	AR		[36,37]
	*DNASE1L3*	AR		[38,39]
	*TREX1*	AD/AR	AGS, SLE	[40,41,42,43,44,45,46,47]
	*IFIH1*	AD	SLE, AGS, FCL, Singleton-Merton syndrome	[48,49,50,51]
	*DDX58*	AD	SMS, glaucoma and skeletal abnormalities	[52,53]
	*ISG15*	AR	basal ganglia calcification, AGS, pseudo-TORCH syndrome, MSMD, SLE	[54,55,56,57]
	*USP18*	AR		
	*SAMHD1*	AD	AGS, SLE, FCL, CLL, deforming arthropathy and recurrent oral ulcers	[58,59,60,61]
	*OTUD1*	AR	SLE, autoimmune disease	[62]
	*ACP5*	AD/AR	SPENCD, SLE, skeletal abnormalities	[63,64,65,66]
	*TMEM173*	AD	SAVI, SLE, FCL, SCID	[67,68,69,70]
	*RNASEH2A*	AR	FCL, AGS, SLE	[71,72]
	*RNASEH2B*			
	*RNASEH2C*			
	*ADAR1*			
Self-tolerance pathway	*PRKCD*	AR	SLE	[73,74,75,76,77]
	*TNFSF13B*		SLE	[78,79]
RAS pathway	*PTPN11* *SOS1* *RAF* *KRAS* *NRAS* *SHOC2*	AD	NS, SLE	[80,81,82,83,84,85,86,87,88,89,90,91]
Other pathways	*RAG1*	AR	SLE, SCID	[92,93]
	*RAG2*	AD		
	*FAS/FASL*	AD	SLE, ALPS	[94,95]
	*TNFAIP3*	AD	SLE, ALPS, BD-like	[96,97,98,99,100]
	*ADA2*	AR	SLE, DADA2	[101,102]
	*IKZF1*	AD	SLE, ITP, CVID	[103,104,105]

SLE: systemic lupus erythematosus, SMS; Singleton–Merten syndrome, HUV: hypocomplementemic urticarial vasculitis, AGS: Aicardi-Goutières syndrome, FCL: familial chilblain lupus, CLL: chronic lymphocytic leukemia, MSMD: mendelian susceptibility to mycobacterial disease, SPENCD: spondyloenchondrodysplasia, SAVI: STING-associated vasculopathy with onset in infancy, SCID: severe combined immune deficiency, ALPS: autoimmune lymphoproliferative syndrome, NS: Noonan-related syndrome, DADA2: deficiency of ADA2, BD: Behçet’s disease, ITP: immune thrombocytopenic purpura, CVID: common variant immunodeficiency disease, AR: autosomal recessive, AD: autosomal dominant.

### 2.1. Complement Pathway

Genetic defects in the complement system are the first and the most frequently described category in the field of monogenic lupus (Table 1). The complement system comprises more than 30 proteins which make up important components of the innate immunity [106]. Complement is crucial for defense against pathogens and for clearance of autologous immune complexes and apoptotic cells. When the complement system is hyperactivated, it drives a severe inflammatory response in numerous organs, particularly in the kidneys. Paradoxically, a complement genetic deficiency or acquired deficiency due to the presence of autoantibodies directed against complement proteins are also linked to SLE. 

Complement components are activated by several sequential enzymatic processes through one of the three pathways: classical, alternative or lectin. Deficiencies of proteins in the early steps of the complement pathway are generally associated with both autoimmune diseases and infections. Conversely, deficiencies in late complement proteins are merely associated with an increase in infection rates [18,106]. 

Defects in early complement proteins (C1q, C1r/C1s, C2 and C4) are thought to trigger autoimmunity as a consequence of impaired opsonization, which is critical for removal of immune complexes and apoptotic bodies by phagocytes, and as result of B lymphocyte activation. The accumulation of autoantigens leads to activation of immune responses and systemic inflammation. In addition, presentation of immune complexes in a complement-dependent manner to the B lymphocytes by the complement receptors CR1(CD35) and CR2 (CD21) on dendritic cells are essential for maintenance and activation of memory B cells [107]. 

C1q has additional functions that go beyond immune surveillance, tolerance and clearance of self and non-self-antigens. C1q inhibits type I IFN and other cytokines production either directly or via interaction with leucocyte associated Ig-like receptor (LAIR)1 on dendritic cells and by inhibition of TLR7 and TLR9 mediated IFN-α production. The main producers of type I IFN are plasmacytoid dendritic cells, while the uptake of immune complexes coated with C1q protein is carried out principally by monocytes [19]. In addition to lupus, infantile-onset Aicardi-Goutières syndrome (AGS) phenotype was also observed in a patient with C1q deficiency [20,21].

Similarly to other monogenic forms of SLE, inherited complement deficiencies may present with early-onset lupus-like manifestations and without gender predilection that is observed in patients with adult-onset lupus [7,25,108]. A systematic analysis of reported cases showed that the strength of the association between complement deficiency and lupus depends on the position of the missing protein in the cascade (C1 > C4 > C2) [109]. The frequency of SLE or SLE-like phenotype is approximately 90% in patients with C1q deficiency, 65% in C1r/C1s deficiency, 75% in C4 deficiency and 10% in C2 deficiency [7,22,23,24,26,27,110,111,112,113,114]. Compared to the polygenic form of SLE, more severe disease course, frequent cutaneous manifestations and a high mortality rate are noted [7,12,18,22,23,24,25,26,114,115]. The association of C1q and C1r deficiency with an upregulation in type I IFN signaling may explain the increased prevalence of SLE in early complement deficient patients [18,112]. The phenotype of patients with early complement deficiencies can also be more severe due to recurrent infections. 

Among the complement proteins there is a unique gene dosage phenomenon for the *C4* gene that is associated with predisposition to SLE. C4 is encoded by two different genes, *C4A* and *C4B*, with a considerable gene copy number variations. Variation in *C4* copy numbers is related to earlier onset and more severe disease course. Although, C4 deficiency is strongly associated with lupus phenotype, complete genetic deficiency is rare. In contrast, the copy number variation (CNV) of C4 genes (*C4A* and *C4B*), which ranges from two to eight copies, is recognized as a crucial factor in prediction of the risk for juvenile-onset SLE [28,29]. Recent population studies consistently showed that the higher the copy number of C4 genes, the lower the risk of non-mendelian SLE, and vice versa [29,30,31,32]. High numbers of auto-reactive B cells, glomerulonephritis and increased levels of autoantibodies were detected in mice with defective *C4* genes [116]. 

### 2.2. Type I Interferon (IFN) Pathway

The upregulation in type 1 IFN pathway can occur through different mechanisms: defect in nucleases activity (TREX1, SAMHD1, ADAR1, RNases, DNases), defect in a negative regulator of IFN signaling (ISG15, USP18) and by constitutive activation of an innate immune sensors (MDA5, RIG-I, STING) (Table 1) (Figure 1). Increased expression of interferon-stimulated genes (ISGs) is the hallmark of these diseases. Why some interferonopathies lead to systemic inflammation/autoimmunity while other present with neurological disease remains unclear.

These discoveries were fundamental as they suggested use of new therapies targeting the interferon pathway. Treatment of patients with the anti-IFN-α therapies has been shown to suppress the interferon signature and disease activity [117,118,119,120,121]. There is also a number of undergoing clinical trials with janus kinase (JAK) inhibitors in patients with SLE that can be found at the ClinicalTrials.gov website (NCT03843125, NCT03978520, NCT03134222, NCT02535689, NCT03288324).

#### 2.2.1. Deoxyribonuclease Deficiencies

Deoxyribonucleases (DNase) are a group of enzymes that catalyze the degradation of DNA molecules and thus serve to prevent recognition of self-DNA. To date, four different DNases have been linked to monogenic lupus: DNase I, DNase1L3, DNase II and TREX1. DNase I is a major serum endonuclease that degrades extracellular dsDNA from dying cells. DNase II is a major lysosomal endonuclease that plays a pivotal role in the degradation of exogenous DNA encountered by endocytosis. Deoxyribonuclease 1 like 3 (DNase1L3) is homologous with DNase I and is presumed to play a role in clearance of neutrophil extracellular traps (NETs) [122].

Recessively inherited loss of function rare/novel mutations in *DNASE I, DNASE II* and *DNASE1L3* genes are associated with a loss of DNase endonuclease activity [33,34,36,38,110,112]. As result, the accumulation of nucleic acid leads to activation of DNA sensors and type I interferon signaling pathway, although an interferon signature has been shown only in patients with DNASE II deficiency. Genetic defects in these three genes have been described as the cause of a mendelian and early-onset lupus that is characterized by presence of antinuclear and anti-double-stranded deoxyribonucleic acid (ANA and anti-dsDNA) antibodies and hypocomplementemia [33,34,36,38] (Table 1). Other features of autoimmunity has been noted in these patients including Sjogren syndrome (ref 48). Observation of an SLE-like phenotype in *Dnase1* and *Dnase1L3*-deficient mice was supportive of these reports [123,124]. Interestingly, *Dnase2* deficient mice were embryonic lethal owing to severe anemia [37]. *DNASE1L3* mutations were identified in two unrelated Turkish families with five affected children presenting with hypocomplementemic urticarial vasculitis (HUV) [39]. It is noteworthy that HUV and SLE share many clinical manifestations. In Korean and Spanish cohort studies, a particular single nucleotide polymorphism (SNP) in exon 8 of *DNASE1*, p. Gln244Arg (rs1053874), was shown to be associated with production of autoantibodies and increased SLE susceptibility. However, this SNP did not correlate with DNase activity in sera of SLE patients and controls [35,125]. Furthermore, the upregulation in interferon gene expression signature has not been demonstrated in any of these patients. Thus, decreased DNase activity due either to the presence of circulation inhibitory factors or loss of function mutations is linked to many autoimmune phenotypes including lupus.

TREX1 (also called DNase III) is major mammalian 3′-5′ DNA endonuclease that resides in the cytosol and acts both on single and double stranded DNA. TREX1 cleaves mismatched and modified nucleotides from DNA 3′ end and degrades DNA derived from retroviruses and retrotransposons [126]. Recognition of these elevated levels of nucleic acids by a cytosolic sensor cyclic GMP-AMP (cGAMP) synthase (cGAS) induces the production of type I IFNs via the stimulator of IFN genes protein (STING) and when excessive can trigger autoimmunity [71,127,128]. Crossing of Trex1-deficient mice with mice heterozygous for mutation in cGAS or STING ameliorates the phenotype [40,129,130].

Heterozygous or recessive loss-of-function mutations in the *TREX1* gene lead to dysfunctional exonuclease activity [8]. Although most patients carry biallelic pathogenic variants in this gene, some mutations, p.Asp200Asn (D200N) and p.Asp18Asn (D18N), are linked to a dominant phenotype. The difference in inheritance type is related to the magnitude of effect of the disease-causing variants on the protein DNase degradation activity. The dominantly inherited mutations affect the catalytic site and DNA binding proficiency of this enzyme [41].

Variable disease expressivity and surprisingly reduced penetrance have been reported in these patients. Cold-induced skin lesions on distal extremities, high titers of multiple autoantibodies and hypergammaglobulinemia are general characteristics of TREX1 deficiency. In addition to these clinical features, infantile-onset neurologic manifestations including encephalopathy with basal ganglia calcifications and white matter lesions, cerebrospinal lymphocytosis mimicking congenital virus infections, which are hallmarks of AGS, hav been also reported in some cases. Progressive neurologic involvement eventually leads to severe physical and intellectual disability in the vast majority of patients [110,112]. 

*TREX1* mutations are detected in about 25% of AGS patients, and in up to 2% of patients with lupus. Among the patients with a TREX1 mutation, 60% of subjects are documented to have at least one of the following lupus features: antibodies to extractable nuclear antigens (ENA), ANA and anti-dsDNA, thrombocytopenia, leukopenia, skin lesions, oral ulcers and arthritis [127]. Over time, with the addition of systemic features to the limited cutaneous manifestations of FCL, 18% of the FCL cases have eventually evolved into SLE [42,112]. Targeted *TREX1* gene analysis in large adult SLE cohorts yielded new cases who carry heterozygous mutations predominantly clustered in the C-terminal protein domain [43,44,45]. Rare variants in the *TREX1* gene were also observed in other autoimmune diseases including Sjogren’s syndrome and systemic sclerosis [46]. 

Genetic counseling can be challenging. In one family, a child with severe neurological features resembling AGS inherited the p.Asp18Asn mutation from her mildly affected mother, while other family members who were carriers for this mutation mostly presented with chilblains [47]. 

#### 2.2.2. SAMHD1

Recessively inherited loss-of-function mutations in SAM (sterile alpha motif) domain and HD (phosphohydrolases named HD after the conserved doublet of predicted catalytic residues) Domain-containing protein 1 (encoded by *SAMHD1* gene) result in increased levels of deoxyribonucleoside triphosphates (dNTPs) by inhibiting the degradation of these DNA precursors [58,110,112]. The excessive presence of dNTP pools, as a result of SAMHD1 mutations, leads to DNA damage, cell cycle arrest and cell death by impairing DNA replication and repair mechanisms. Acquired (somatic) SAMHD1 mutations are found in patients with chronic lymphocytic leukemia, CLL (OMIM 151400) [59]. Furthermore, accrual of cytosolic dNTPs is sensed as danger molecules, which in turn stimulates type I IFN pathway [58]. Pathogenic variants in the *SAMHD1* gene have been implicated in the pathogenesis of three different clinical disorders including AGS, SLE and FCL, similar to *TREX1* mutations [60,71,131,132]. A distinct phenotype with deforming arthropathy and recurrent oral ulcers in addition to the classical neurologic and skin manifestations, has also been described [61].

#### 2.2.3. Cytosolic RNA Nucleic Acid Degradation/Editing Pathways

Biallelic loss of function mutations in ribonucleases RNASEH2A, RNASEH2B, RNASEH2C, and RNA-specific adenosine deaminase 1 (ADAR1) cause spectrum of phenotypes with varying degrees of severity: mildest familial chilblain lupus (FCL), moderate severity systemic lupus, and most severe Aicardi–Goutières syndrome (AGS1-7: OMIM 225750, 610181, 610329, 610333, 612952, 615010, 615846) [9,71,72,110,112,113]. Their deficiency result in accumulation of RNA/DNA hybrids or RNA molecules that ultimately leads to an excessive type I IFN signaling. Occasionally, the disease with a uniform phenotype due to the aforementioned mutations evolves over time into one with overlapping features of AGS, FCL and SLE [42,127,128]. For instance, in an Italian study, more than half of the patients with genetically confirmed AGS were found to have autoantibodies despite lack of any autoimmunity related clinical manifestations [133]. Presence of autoantibodies against the nuclear antigens and the endothelial cells and astrocytes in the brain was demonstrated in another multi-center study [134]. 

#### 2.2.4. Nucleic Acid Sensing Pathways

Melanoma differentiation-associated protein 5 (MDA5) and retinoic acid-inducible gene (RIG-I) are cytoplasmic double stranded ribonucleic acid (ds-RNA) sensors that play an essential role in the triggering of antiviral responses [135]. Their expression is typically low in resting cells and is greatly increased upon exposure to viral infection. Autoactivation of these sensors due to pathogenic mutations in the helicase domains of respective proteins leads to various autoimmune diseases.

Heterozygous missense gain of function mutations in the *IFIH1* gene (encoding MDA5), account for several disease phenotypes including SLE, AGS, FCL and Singleton–Merten syndrome (SMS; OMIM 182250) [48,49,50,71]. Constitutive activation of MDA5 receptor leads to excessive type I IFN signaling as demonstrated by high expression of ISGs [49,52]. Reduced penetrance mutations have been reported in some families, which suggest that other factors such as viral exposure contribute to the disease expression [51]. Whole-exome sequencing (WES) revealed a de novo *IFIH1* mutation, p.Arg779His, in a 16-year-old Belgian girl presenting with severe early-onset SLE, immunoglobulin (Ig) A deficiency and lower limb spasticity [50]. Similarly, severe lupus-like disease including nephritis, high ANA and anti-ds DNA autoantibodies, developed in mice with a gain of function mutation in MDA5. Sustained upregulation of IFN and ISGs in the activated dendritic cells and macrophages elicited antibody production by B cells and plasma cells [52]. The association of a SNP in IFIH1, p.Ala946Thr (rs1990760), with anti-ds DNA antibodies was shown in a cohort of 563 SLE patients and correlated with increased ISG expression in an anti-ds DNA positive subgroup [136]. This IFIH1 variant was demonstrated to increase susceptibility not only to SLE but also to type I diabetes mellitus, multiple sclerosis and rheumatoid arthritis in a meta-analysis [137].

Heterozygous mutations in RIG-I (encoded by the *DDX58* gene) have been identified in a family with atypical SMS and in patients with glaucoma and skeletal abnormalities [53]. Both mutations lead to increased IFN activity. Why patients with SMS in contrast to patients with other types of interferonopathies develop skeletal and dental abnormalities and aortic calcifications remains unclear. Rig-I deficient mice were viable and developed colitis-like phenotype due to T cell activation, which shows that RIG-I has a function in maintenance of T cell homeostasis [138]. 

Another type I interferonopathy that can manifest with SLE-like phenotype is STING-associated vasculopathy with onset in infancy (SAVI) (OMIM 615934) [67,129]. The STING, a transmembrane protein residing on endoplasmic reticulum, is an essential and common sensor of self-DNA molecules. In normal circumstances, activation of STING by its ligand, cGAMP, leads to expression of type I IFN through downstream pathways including a protein kinase TBK1 and IRF-3 [139]. Heterozygous pathogenic missense mutations in *TMEM173*, which encodes STING, result in constitutively active STING [67,129]. Mutations that reside in exon 5, which encodes a highly conserved dimerization domain, act by reinforcing the stability of STING dimers and lead to ligand-independent activation of the TBK/IRF3 pathway. Patients with a most severe phenotype described as SAVI present with early-onset systemic inflammation, interstitial lung disease, and cutaneous manifestations including malar rash, erythematous plaques and nodules, cold-induced chilblain-like lesions that can evolve into ulcers and peripheral amputations [129]. Anti-phospholipid antibodies were present in five out of six patients and low-titer ANA was detected in half of the children. Although the aforementioned features mimic SLE, the authors reported that none of the subjects met the SLE criteria [129]. Interestingly, mice who carry one of the SAVI-associated mutations developed features of severe combined immunodeficiency [68,140].

STING-associated diseases have variable expressivity. In a family with four affected patients, one had been diagnosed with infantile-onset SLE, while the other three family members had a disease onset after 12 years of age [67]. Classical SAVI phenotype including recurrent fever episodes, malar rash and interstitial lung disease were observed in three family members. However, a 65-year-old family member exhibited a significantly different phenotype with marked failure to thrive and articular manifestations. All subjects had elevated acute phase reactants and were positive for rheumatoid factor (RF) and ANA with various titers, whereas only the index case had positive anti-dsDNA. More recently, a heterozygous *TMEM173* mutation, p.Gly166Glu, was identified in a Greek family with five members affected with familial chilblain lupus (FCL) [69]. Interestingly, this mutation resides in the dimerization domain of STING like SAVI-associated causal variants, however other than cold-induced chilblain lesions, none of the patients had fevers and lung disease. Similar to previous case-series, ANA positivity was detected in four out of five family members [69]. Some patients presented with lymphopenia of T and NK cells and mild immunodeficiency, which is consistent with the role of STING in lymphoid development [70]. Such variability in the disease expression might be explained by presence of modifying alleles in *TMEM173* and other genes. 

### 2.3. Negative Regulators of Type I Interferon Signaling Pathway 

#### 2.3.1. ISG15 and USP18

The ISG15 is a type I IFN inducible ubiquitin-like protein that stabilizes the levels of ubiquitin-specific peptidase 18 (USP18), a potent negative regulator of IFN-α/β signaling [141]. In the absence of ISG15, USP18 is degraded via a proteasome, which permits signaling through the IFN alpha receptor (IFNAR) and leads to increased antiviral activity [54,55,141]. Biallelic recessively inherited mutations in these two genes, ISG15 and USP18, result in enhanced type I IFN response (Table 1, Figure 1). Patients present with a wide spectrum of neurological and immunological manifestations, including basal ganglia calcification, AGS, pseudo-TORCH syndrome (OMIM 617397), seizures and mendelian susceptibility to mycobacterial disease (MSMD) (OMIM 209950) [54,55,141]. The MSMD immunodeficiency is characterized by susceptibility to infection with weakly virulent mycobacteria as a result of insufficient ISG15-dependent IFN-γ production [55,141]. Although higher autoantibody levels were identified in patients with ISG-15 deficiency compared to the age-matched controls, there are no reports of lupus patient carrying ISG15 or USP18 mutations [55]. However, significantly higher levels of ISG15 in SLE patients compared to healthy controls and other disease controls suggested that ISG15 could be a possible diagnostic marker for SLE and other interferonopathies [56,57]. Moreover, the ISG15 levels were found to correlate with disease activity in SLE patients [57].

#### 2.3.2. OTUD1

OTUD1 gene encodes a widely expressed deubiquitinase that plays a role in the regulation of IRF3 transcriptional activity. OTUD1 interacts with IRF3 to remove the K63-linked poly-ubiquitin chains on IRF3 and to suppress the interferon activity. Patients with loss-of-function missense mutations in OTUD1 present with an autoimmune disease, including some patients diagnosed with SLE [62]. These mutations may abolish the interaction with IRF3 or may act by reducing OTUD1 deubiquitinase activity. 

#### 2.3.3. ACP5

Spondyloenchondrodysplasia (SPENCD; OMIM 607944) is a disorder very similar to the AGS with regard to neurologic manifestations (intracranial calcifications, spasticity) and autoimmunity (frequent SLE manifestations). However, skeletal abnormalities such as short stature, platyspondyly and enchondromatosis are unique features related to SPENCD [63,64,65,110,112]. Autosomal recessive mutations in the *ACP5* gene results in a deficiency of tartrate-resistant acid phosphatase (TRAP) enzyme, which is expressed in osteoclasts and myeloid cells. TRAP is required for processing and/or degradation of osteopontin (OPN) in plasmacytoid dendritic cells (pDCs) by catalyzing the dephosphorylation of OPN. Increased activity of OPN, when in a phosphorylated state, continuously stimulates the TLR9 signaling pathway. This activation results in nuclear translocation of IRF7 and NFkB, which in turn increases the expression of type I IFN, IL-6 and TNF [66]. Increased OPN levels seem to trigger type I IFN production and autoimmunity and are responsible for the skeletal dysplasia [63,64,65,66]. Virtually, all SPENCD patients were found to be positive for anti-dsDNA and/or ANA autoantibodies, whereas only half of them met the ACR 1997 criteria for SLE [64,65]. Heterozygous missense *ACP5* variants were detected with an increased frequency in a cohort of non-mendelian SLE, suggesting that the TRAP deficiency may increase the susceptibility to SLE [66]. 

### 2.4. Self-Tolerance Pathway

B lymphocytes play a key role in development of autoimmunity more than other cell types of the innate and adaptive immune system. Failure in the elimination of self-reactive cells during the development process of B cells, also called inadequate negative selection, may lead to loss of tolerance and development of autoimmunity (Table 1, Figure 1). Autoimmune processes can be further halted by deleting the B lymphocytes with B cell receptors (BCR) recognizing self-antigens at various checkpoints.

#### 2.4.1. PRKCD

Protein kinase C-δ (PKC-δ) encoded by *PRKCD* gene, has been implicated in the regulation of apoptosis, survival and proliferation of lymphocytes [142]. Lupus-like manifestations including glomerulonephritis, lymphadenopathy, splenomegaly and positive-titers of autoantibodies developed in *PRKCD* knockout mice [143]. PKCδ-deficient mice showed an impaired negative selection of self-reactive B cells, which leads to uncontrolled B cell proliferation in peripheral lymphoid tissues. The hyperproliferative phenotype was B-cell autonomous. Homozygous missense mutation in *PRKCD* was identified in three siblings from consanguineous parents, all of whom met the ACR criteria for SLE [73]. Decreased expression and enzymatic activity of PKC-δ, impaired apoptosis and expansion of immature B cells were demonstrated. Typical cutaneous lupus rashes, lupus nephritis, positive ANA, positive anti-ds DNA were observed in all of three patients. Lymphadenopathy, hepatomegaly or splenomegaly, which are suggestive of a lymphoproliferative disease, were detected in two out of three patients. Subsequently, four other children with early-onset SLE-like manifestations including lupus-like rash, high-titers of various autoantibodies, nephritis or cytopenia were also later described [74,75,76]. A lymphoproliferative disorder manifested by hepatomegaly and/or splenomegaly and/or lymphadenopathy was observed in all of these children. An infantile-onset case with overlapping features of primary immunodeficiency due to a B-cell deficiency and autoimmunity including membranous glomerulonephritis, presence of ANA, anti-ds DNA and anti-phospholipid antibodies has been also reported [77].

#### 2.4.2. TNFSF13B/BAFF

Overexpression of the cytokine and drug target B-cell activating factor (BAFF; encoded by *TNFSF13B*) has been associated with susceptibility to SLE, multiple sclerosis and rheumatoid arthritis. The functional causal variant denoted as BAFF-var (a combination of rs374039502 and an insertion-deletion variant GCTGT→A [rs200748895]) has been initially identified by Genome Wide Association study (GWAS) in the Sardinian population and recently replicated in other cohorts of patients [78,79,144]. The BAFF-var is more frequent among Sardinians (MAF = 33%) than other populations (Europeans MAF = 3–5%) and was shown to yield a shorter transcript that escapes microRNA inhibition. As result there is increased production of soluble BAFF and a higher number of B lymphocytes and immunoglobulins, predisposing to autoimmunity. High BAFF expression is related to active disease, renal and hematological involvement in SLE patients [145].

Although belimumab, which targets BAFF, has been approved for the treatment of SLE by the Food and Drug Administration this variant was not associated with belimumab efficacy as measured by the primary efficacy end point SRI4, which integrates three validated lupus instruments measuring disease activity. The study is likely underpowered to detect an association for a minor allele frequency of 5% or less [146].

### 2.5. Other Pathways Associted with Susceptibility to Lupus-like Phenotypes

#### 2.5.1. RAG1 and RAG2

Primary immunodeficiencies with autoimmune features have been increasingly recognized in the last decade *(*Table 1, Figure 1*)*. Severe combined immunodeficiency with lack of T and B lymphocytes (SCID; OMIM 601457) is caused by recessively inherited mutations in recombinase activating gene *(RAG) 1* and *2.* Patients have profound lymphopenia and diminished or absent immunoglobulins and present with recurrent opportunistic infections. A growing body of literature demonstrates that pathogenic variants in heterozygous state, may contribute to generation of autoantibodies in RAG1-related disease. Compound heterozygous *RAG1* mutations have been identified in a family with the combined immunodeficiency phenotype, autoimmune cytopenia and multiple autoantibodies, including anti-IFNα antibodies. The mechanism for generation of these autoantibodies is unclear, and may be attributed to underdeveloped thymus with decreased *AIRE* expression [92,93]. 

#### 2.5.2. FAS/FASL

The cell surface death receptor (FAS) belongs to the TNF receptor superfamily (TNFRS). Interaction of the FAS receptor with its ligand FASL plays a central role in programmed cell death, also called apoptosis. Autosomal dominant mutations of *FAS* or *FASL* genes result in decreased expression of FAS and FASL proteins and failure to remove autoreactive cells [110,147]. Defective FAS-FASL apoptotic pathway leads to a lymphoproliferative syndrome with autoimmune features (autoimmune lymphoproliferative syndrome; ALPS; OMIM 601859). Enlargement of the lymph nodes, spleen and/or liver with autoimmune cytopenia are the main features of the disease in most cases. In a cohort of 75 patients with SLE, a heterozygous 84-bp in frame deletion in the *FASL* gene was detected in a 64-year-old male [94]. His SLE manifestations include malar rash, arthritis, serositis, renal disease, leukopenia, anti-DNA antibodies, and a positive ANA with generalized lymphadenopathy. Polymorphisms in FAS and FASL genes were shown to increase SLE susceptibility [95,147]. 

#### 2.5.3. RASopathies

RASopathies, which are caused by dominantly inherited mutations in several genes (KRAS, NRAS, PTPN11, RAF, SHOC2 and SOS1), are a group of neurodevelopmental disorders including Noonan syndrome (NS; OMIM 163950) and Noonan-related syndromes (NS2-11; OMIM 605275, 609942, 610733, 611553, 613224, 613706, 615355, 616559, 616564, 618499). Beside classical phenotype patients with these disorders may present with features of SLE, autoimmune thyroiditis, vitiligo, celiac disease and Degos [79,145]. The affected proteins participate in the RAS/MAPK pathway that is involved in regulation of different cellular processes by transduction of growth factor signals [145]. Shared cardinal characteristics of the RASopathies include short stature, craniofacial malformations, webbed neck, cardiac malformation, variable cognitive delay and an increased risk of cancer development [80,81]. Twelve patients with NS/Noonan-related syndrome phenotype and SLE characteristics have been described to date [80,81,82,83,84,85,86,87,88,89,90,91]. Of these 12 patients, four had *SHOC2* mutations, two had *KRAS* and one had a *PTPN11* mutation [80,81,84,85,86,90,91]. The remaining five patients with RASopathy and SLE are molecularly undiagnosed [82,83,87,88,89]. Overall, the most frequently reported manifestation associated with SLE was arthritis (*n* = 8/11), followed by pericarditis/pleuritis (*n* = 7/11), autoimmune cytopenia (6/11), and skin involvement (*n* = 2/8). Compared to the classical SLE, male preponderance (F:M = 1/2), a lower rate of skin involvement and a higher rate of serositis are noteworthy in patients with RASopathies.

#### 2.5.4. RELopathies

Patients with mutations in genes that regulate canonical NF-kB pathway can also present with SLE and other autoimmune phenotypes. The A20 protein, encoded by the *TNFAIP3* gene, is a key negative regulator of NF-kB, NLRP3 inflammasome and other immune signaling pathways. Dominantly inherited loss-of-function mutations in *TNFAIP3* lead to early-onset autoinflammatory disease haploinsufficiency of A20 (HA20; OMIM 616744). A patient with p.Phe224Ser*fs**4 mutation was initially diagnosed with SLE, including central nervous system (CNS) vasculitis [96,97]. Another study showed that patients with HA20 gradually progress to develop strong autoimmune features [98]. The autoimmune features may be attributed to a Type I interferon signature detected in peripheral blood of these patients among other highly elevated proinflammatory cytokines [99]. Haploinsufficiency of A20 was also identified in patients with childhood-onset autoimmune diseases and autoimmune lymphoproliferative syndrome (ALPS; OMIM 601859) [100].

#### 2.5.5. Deficiency of ADA2 (DADA2)

DADA2 (OMIM 615688) is caused by biallelic loss of function mutations in adenosine deaminase 2 (ADA2) gene, with a function of extracellular deaminase and growth factor. DADA2 patients have broad disease spectrum including systemic vasculitis, strokes, livedoid rash, early-onset polyarteritis nodosa (PAN), CVID and bone marrow failure. SLE-like phenotype and an elevated interferon gene expression signature has been reported in same cases [101,102].

#### 2.5.6. IKZF1

*IKZF1* gene encodes the IKAROS protein, which is an important transcriptional factor in the process of hematopoiesis. Heterozygous germline mutations in *IKZF1* cause abnormal hematopoiesis, combined variable immunodeficiency (CVID) and features of autoimmunity. SLE-like and immune thrombocytopenic purpura (ITP) phenotypes have been reported in some patients [103,104]. A genome-wide association study showed that the variant rs1456896 in the 5′ UTR of *IKZF1* was relevant to lupus nephritis [105].

## 3. Conclusions

In the light of these observations, genetic analysis for monogenic lupus should be warranted in patients with lupus phenotype and at least one of the following clinical features: early-onset manifestations (<5 years), parental consanguinity, presence of family history for autoimmune diseases in a first-degree relative. Whether all the genes discussed in this review should be included in the targeted gene sequencing panel is questionable, however considering a continuum of clinical features in patients with immune dysregulation and reasonable cost of NGS based sequencing this is feasible. Molecular diagnosis has become increasable valuable to guide patient management from diagnosis to treatment strategies and has helped to define a spectrum of phenotypes linked to pathogenic variants in a single gene.

Complement deficiencies are the most frequent subcategory of monogenic lupus. The second leading cause of monogenic lupus are primary intereferonopathies. The frequency of clinical manifestations in patients with monogenic SLE differs from multifactorial/polygenic SLE, with the majority of cases having CNS disease and cutaneous manifestations, particularly chilblain lesions. Due to the overlapping clinical manifestations, a multidisciplinary team approach including a rheumatologist, dermatologist, neurologist and immunologist, is required for appropriate diagnosis and management of these genetically and clinically heterogeneous diseases.

Given the complexity of regulation in the interferon signaling pathway, the recognition of novel mendelian diseases of monogenic lupus will likely increase with the aid of whole exome (WES) and whole genome (WGS) sequencing. These reverse genetic approaches will lead to identification of novel genes linked to SLE and other autoimmune diseases, which will pave the way for the development of new therapeutic options. Revisiting existing diagnostic criteria will be needed for the classification of monogenic lupus and type I interferonopathies after elucidation of these molecular pathways and recognition of overlapping manifestations. Revealing all pathogenetic pathways and recognizing the clinical spectrum of monogenic SLE will not provide tailored therapy options only for this type of inherited lupus but also to the classical multifactorial form of the SLE.

## Figures and Tables

**Figure 1 jcm-09-00712-f001:**
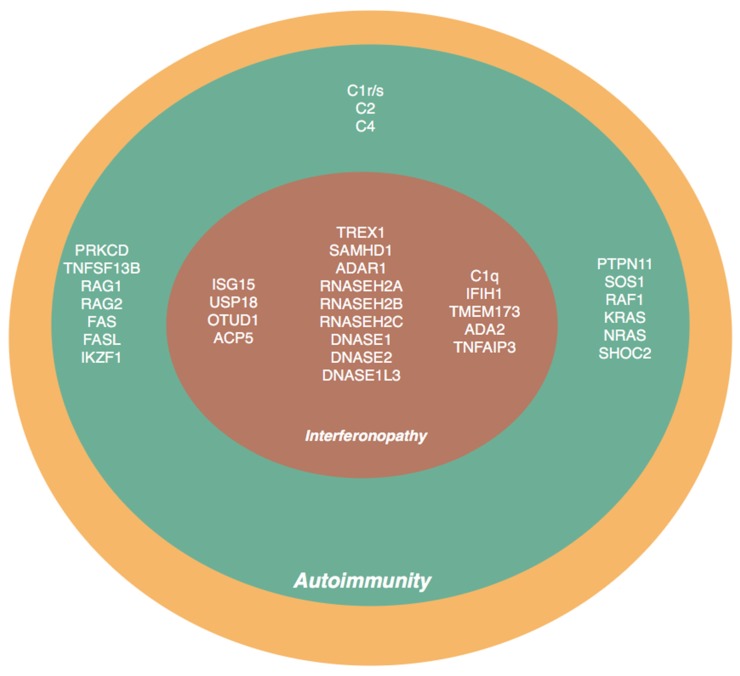
Diagram of genes associated with monogenic systemic lupus erythematosus (SLE) and SLE-like features. The depicted genes carry novel or rare pathogenic variants that have been described in patients presenting with lupus or have either clinical or biochemical features of autoimmunity. The inner diagram includes genes associated with an upregulated type I interferon and most of these diseases are considered primary interferonopathies. The outer diagram depicts the genes that, when mutated, are associated with a spectrum of immune dysregulations and patients can manifest both features of immunodeficiency and autoimmunity. Typically, the autoimmune phenotype is milder than in patients with primary interferonopathies and there is no evidence of enhanced type I interferon signaling.

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
