# Peer review of "New Horizons in the Genetic Etiology of Systemic Lupus Erythematosus and Lupus-Like Disease: Monogenic Lupus and Beyond"

_jcm, 2020, doi:10.3390/jcm9030712_

Round 1

Reviewer 1 Report

The manuscript by Demirkaya, et al seeks to shed light on how genetic mutations are associated with systemic lupus erythematosus and other forms of lupus. This is a worthy goal and likely to contribute important information to the field. However, in its current form, this review has a number of errors and fallacies that must be addressed before publication.

The most serious error is the title and the theme of the article itself. A monogenic disease, by definition, is one which is CAUSED by a single gene. While autoimmune lupus-like diseases are evident in mice, and can be "caused" by SOME genes listed here, this does not mean that these genes are monogenic for lupus. They are likely to be predisposing, but not causal, and certainly not proven to be monogenic. Readers will be misled, and this must be better explained and tempered. Indeed, the authors do a lovely job, in many  cases, of discussing the literature and the fact that these mutations are linked to multiple forms of lupus and other diseases as well. Thus, the title is also misleading as it states everything is related to SLE. The abstract, the introduction, and the conclusions all need to be modified to reflect these facts. 

Other comments are listed in the entire review attached as a pdf.

Reviewer 2 Report

This review focusses on mongenic causes of systemic lupus erythematosus (SLE) and SLE-like diseases and tries to categorize the pathogenic mechanisms. I think this is an interesting approach although I have some comments:

1. The title is partially misleading as the main focus of the authors is on IFN type I signature. Some of the deficiencies that were listed have no clear association with SLE (RIG-I, ISG15, USP18, DADA2). Vice versa, other known genetic associations have not been mentioned, e.g. deficiency of ficolin H and (less clear) complement mannose binding lectin. Another example would be variant BAFF (see Steri M. et al., New Engl J Med 2017; 376: 1615) that could be part of chapter 2.4.
2. The abstract is, in my opinion, rather unspecific while I like the more clear conclusions of the review. The authors might want to modify the abstract accordingly.
3. With regard to the strong association between deficiency of complement C1q and SLE (as outlined in detail) the authors cite a number of references. Considering the focus of the review (in particular in view of the conclusions) this review might be helpful: Stegert M. et al. Mol Immunol 2015; 67: 3.
4. The sentence on reference 42 does not really fit as the ref is dealing with a (secondary) complication of SLE and not with a basic mechanism leading to SLE.

Round 2

Reviewer 2 Report

No further concerns